# Registration of a Quadrupole Transition in High-Energy $^{88}$Sr$^+$ Ions Obtained by Laser Ablation Method

Evgeny Telnov [1,*] , Petr Borisyuk [1], Dmitry Tregubov [1,2] , Daniil Provorchenko [2] , Konstantin Trichev [1] and Pavel Cherepanov [1]

1  Department of Metrology Physical and Technical Problems, National Research Nuclear University MEPhI, 115409 Moscow, Russia; pvborisyuk@mephi.ru (P.B.); treg.dim@gmail.com (D.T.); dlaw4dfsdwlwhlfhjinx@yandex.ru (K.T.); winterman404@gmail.com (P.C.)
2  Lebedev Physical Institute of the Russian Academy of Sciences, 119991 Moscow, Russia; provorchenko.di@phystech.edu
*  Correspondence: zhenyatelnov@mail.ru

**Abstract:** In this paper, we demonstrate the interaction of 674 nm laser radiation with a clock quadrupole transition in high-energy $^{88}$Sr$^+$ ions obtained by laser ablation. The results of the spectrometry of the clock and the pump transitions are presented. We describe the parameters of the experimental setup and the protocol of the clock transition spectroscopy and analyze various line broadening mechanisms.

**Keywords:** laser cooling; strontium; ion trap; strontium frequency standard

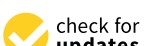



## 1. Introduction

In modern metrology, measured physical quantities are often related to the measurements of time and frequency. The high accuracy of such measurements has contributed to the recent redefinition of the key SI units over time [1]. At the same time, the international time standard is the cesium atomic clock, so that the currently accepted definition of a second is related to the period of radiation, which corresponds to the electronic transition between two sublevels of the hyperfine structure of the ground state of an unperturbed $^{133}$Cs atom. The best relative inaccuracy of such clocks, achieved on the "cesium fountain" installation, is $2 \times 10^{-16}$ [2]. In recent years, achievements in the field of ultrahigh-resolution laser spectroscopy of atomic transitions have opened up real possibilities for significantly increasing the accuracy by several orders of magnitude, as well as improving the stability and reproducibility of time and frequency units [3–6]. The motivation for creating high-precision frequency standards of a new generation is that such clocks are needed not only for applied tasks, but also for basic research. One example of their use is problems in the field of general relativity that were updated after the registration of gravitational waves [7,8], or for the precision determination of fundamental physical constants [9]. Improved clock accuracy can be achieved by increasing the frequency of the atomic oscillators used; i.e., by replacing the microwave operating transition with an optical one, the frequency of which is five orders of magnitude higher. Currently, optical clocks are usually based on forbidden optical transitions in neutral atoms that are localized in an optical lattice with a magic wavelength [10–17] and in ions [18–21], which are trapped and form an ion crystal cooled to a temperature $\leq 10^{-5}$ K. One of the directions in the field of optical frequency standards is the quadrupole transition $5S_{1/2} - 4D_{5/2}$ in a singly charged $^{88}$Sr ion. The relative inaccuracy in such clocks is about $10^{-17}$ [22]. The implementation of an optical time and frequency standard based on the $^{88}$Sr$^+$ ion requires solving a number of problems, which include the development of a vacuum system with the ability to obtain a vacuum of up to $10^{-11}$ Torr, development of an ion generation system, development of a mass spectrometry system and ion retention in a geometrically limited region for a long time,

development of a laser cooling system based on Lambda schemes $5S_{1/2} - 5P_{1/2} - 4D_{3/2}$, and clock transition spectroscopy $5S_{1/2} - 4D_{5/2}$ [23].

The traditional approach for spectroscopy of narrow transitions is deep laser cooling. After obtaining the Wigner crystal, it is still necessary to take into account the Zeeman and Stark splitting, the minimization of secular motion, and the reduction in line broadening by various mechanisms.

This paper presents the results of studying the interaction of 674 nm laser radiation with the forbidden quadrupole transition $5S_{1/2} - 4D_{5/2}$ in hot $^{88}$Sr$^+$ ions, as well as the technique and results of the performed spectroscopy.

## 2. Materials and Methods

Ion production is performed by laser ablation. This approach allows one to work in the same way with different elements, while at the same time with a relatively low material consumption compared to using an atomic oven. The sample is positioned in front of an input aperture of the linear quadrupole Paul trap. During ablation, the plasma scatters directionally, mainly along the normal to the surface of the material. The unevenness of the surface leads to a spread in the directions of flight, which increases as the sample is being used. The geometric spread is compensated by collimation of the scattering plasma into the trap entrance aperture by a pulsed magnetic field of the order of 1 T [24]. Laser ablation of the target generates a plasma torch containing about $\sim 10^{14}$ particles [25]. For the laser ablation, we used a lamp-pumped Nd:YAG laser operating in the Q-factor modulation mode with the following parameters: the laser pulse duration is $\sim 10$ ns; the pulse energy is $\sim 50$ mJ, and the radius of the focused laser spot on the sample is $\sim 100$ µm. Thus, the power density on the samples is $\sim 10 \frac{GW}{sm^2}$. Subsequent steps such as mass spectrometry and ion capture and retention are implemented in a five-section quadrupolee ion trap with a linear configuration [26–28]. All sections in the trap are separated by a silicon insulator, so that a potential can be formed along the ion span, which will maximize the number of trapped ions. Ion retention along the trap axis is carried out by applying a potential to two pins, installed between the quadrupole electrodes. The distance between the rings is 28 mm. Changing the delay time between the laser ablation pulse and the potential transfer to the rings makes it possible to measure the time-of-flight characteristics of the required ions in the plasmas. The expansion energy characteristics of 88 Sr$^+$ ions obtained by this method are in the order of $\sim 10$ eV. The state of the system and its parameters are described in more detail in one of our previous works [29].

The part of the vacuum chamber in which the ion quadrupole trap is located has 6 CF40 windows in the horizontal plane, which provide optical access to the trap, and there is also a CF100 window that allows laser radiation to be introduced through ring electrodes along the quadrupole axis (Figure 1a, not to scale, and the size of the trap is enlarged for clarity). The optical path of 422, 1092, and 674 nm laser radiation inside the trap is shown in Figure 1b. The choice of such configuration was based on the assumption that the introduction of 422 nm cooling laser radiation along the trap axis would lead to the beginning of ion cooling at the ion reversal points near the ring electrodes and to a larger volume of the part of the cloud that interacts with the radiation. This is clearly demonstrated in Figure 1c. From left to right, it shows the images of the ion cloud with an increase in the retention time and, accordingly, cooling; the camera exposure is of the order of 100 ms. The optical path and the choice of the polarization plane of the 1092 nm laser radiation are caused by the destabilization of the "dark states" corresponding to the sublevels $m = \pm\frac{3}{2}$ for $4D_{3/2}$ in the external laboratory magnetic field. The optical output of 674 nm radiation is due to the convenience of detecting traces of the experimental protocol (which will be discussed in more detail later), as well as the availability of optical access to the ion retention zone.

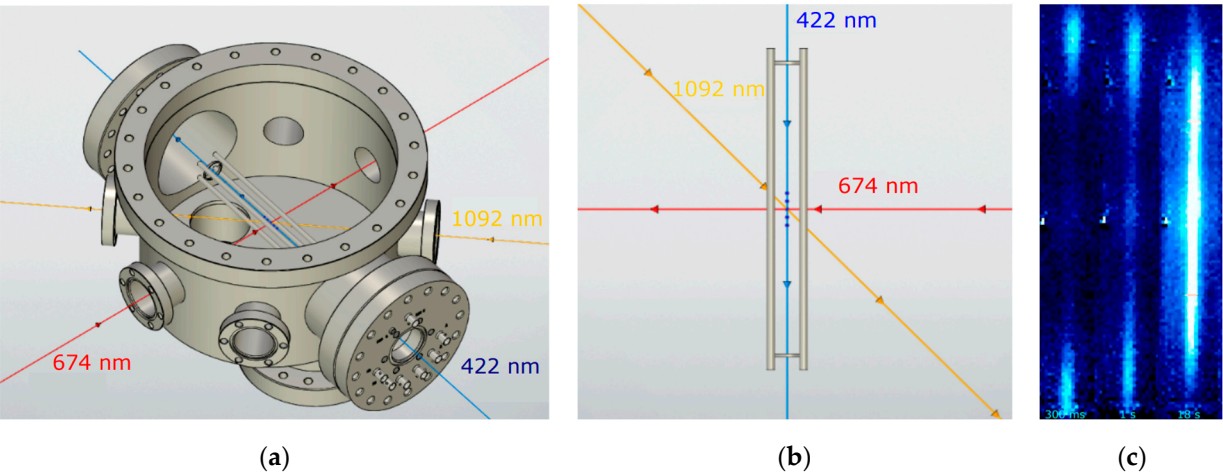

(**a**)               (**b**)               (**c**)

**Figure 1.** (**a**) Part of the vacuum chamber containing the quadrupole Paul trap and optical laser inputs, not to scale, and the trap size is enlarged for clarity (**b**) Optical path of laser radiation 422, 1092, and 674 nm inside the trap (**c**) The images of the ion cloud at different cooling times; exposure cameras take 100 ms.

We used lasers at wavelengths of 422 nm and 694 nm with a linewidth of $\sim$ 1 MHz (made by VitaWave, Moscow, Russia) and at a wavelength of 1092 nm with a linewidth of $\sim$ 0.1 MHz (made by TimeBase, Düsseldorf, Germany). Long-term (time constant of the order of 1 s) frequency stabilization of each laser was performed using a Bristol 871A wavemeter (made by Bristol Instruments, Victor, NY, USA) (relative measurement error is $0.2 \times 10^{-6}$). In the implemented system, laser radiation has the following parameters:

### 3. Energy Structure of the $^{88}$Sr$^+$ Ion

Figure 2 shows a diagram of the energy levels of the $^{88}$Sr$^+$ ion with electronic transitions used for laser cooling, as well as a clock quadrupole transition. For the laser cooling of the $^{88}$Sr$^+$ ions, we used $5S_{1/2} - 5P_{1/2}$ transition at a wavelength of $\lambda$ = 422 nm with a natural linewidth $\gamma_{5S_{1/2} - 5P_{1/2}}$ = 20.4 MHz (further $\gamma_{422}$) [23]. This transition is not cyclic: spontaneous decay is possible from the upper level to the $4D_{3/2}$ level (the ratio of the decay probabilities to the $5S_{1/2}$ and $4D_{3/2}$ levels is 13:1), which leads to ions leaving the cooling cycle. The repumping $4D_{3/2} - 5P_{1/2}$ transition at a wavelength of $\lambda$ = 1092 nm with a natural linewidth of $\gamma_{4D_{3/2}-5P_{1/2}}$ = 1.2 MHz (further $\gamma_{1092}$) is used to return ions in the cooling cycle. The suppressed quadrupole transition $5S_{1/2} - 4D_{5/2}$ is proposed as a clock transition due to its small natural linewidth of $\gamma_{5P_{1/2}-4D_{5/2}}$ = 0.4 Hz (further $\gamma_{674}$).

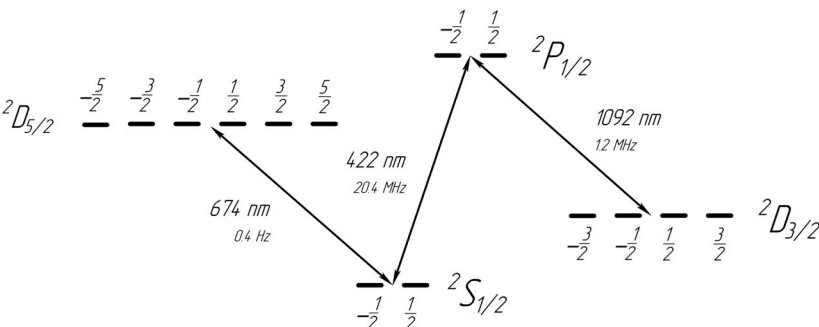

**Figure 2.** Energy levels of the $^{88}$Sr$^+$ ion. Doppler cooling is performing at the 422 nm transition, and the 1092 nm transition is used as a repumping. The spectrally narrow 674 nm transition is used as a clock transition.

In order to perform the spectroscopy of the clock transition $5S_{1/2} - 4D_{5/2}$, we used the luminescence of an ion cloud at a wavelength of 422 nm as the signal. In the presence of

laser radiation at 674 nm, some ions were lost from cooling cycle. This process leads to a decrease in luminescence at a wavelength of 422 nm. This effect can be demonstrated by the example of the two energy structures in Figure 3.

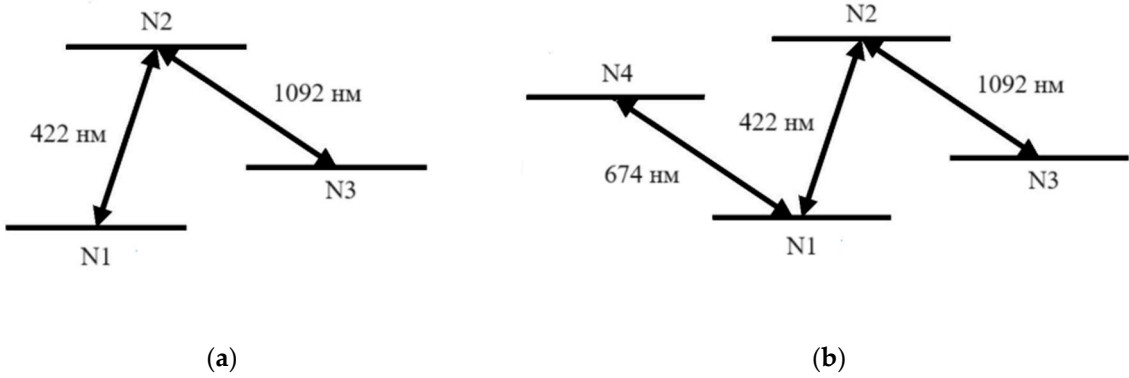

(**a**)                                                                                    (**b**)

**Figure 3.** (**a**) Three-level energy structure and (**b**) four-level energy structure.

The impact of resonant laser radiation on the system will lead to a change in the population of the levels. Consider the three-level system shown in Figure 3a, where the populations of the levels are denoted $N_1$, $N_2$, and $N_3$. A qualitative description of this process can be implemented in the form of kinetic equations.

$$\frac{dN_1}{dt} = \frac{I_{422}\lambda_{422}^3}{2\pi hc}(N_2 - N_1) + 2\pi\gamma_{422}N_2 \tag{1}$$

$$\frac{dN_2}{dt} = \frac{I_{422}\lambda_{422}^3}{2\pi hc}(N_1 - N_2) + \frac{I_{1092}\lambda_{1092}^3}{2\pi hc}(N_3 - N_2) - 2\pi(\gamma_{422} + \gamma_{1092})N_2 \tag{2}$$

$$\frac{dN_3}{dt} = \frac{I_{1092}\lambda_{1092}^3}{2\pi hc}(N_2 - N_3) + 2\pi\gamma_{1092}N_2 \tag{3}$$

where $I_{422}$ and $I_{1092}$ are the radiation intensities for the wavelengths 422 nm and 1092 nm $\left(\frac{W}{m^2}\right)$ (Table 1), respectively; $\lambda_{422}$ and $\lambda_{1092}$ are the wavelengths 422 nm and 1092 nm, respectively; and $\gamma_{422}$ and $\gamma_{1092}$ are the spectral widths of transitions $5S_{1/2} - 5P_{1/2}$ and $5P_{1/2} - 4D_{3/2}$(Hz), respectively. In our experiment, we detected only stationary distribution, which can be estimated from Equations (1)–(3) with $\frac{dN_i}{dt} = 0$: $N_1 \approx 0.362082$, $N_2 \approx 0.318957$, and $N_3 \approx 0.318960$.

**Table 1.** Parameters of laser radiation at 422, 1092, and 674 nm wavelengths.

| Wavelength, nm | Laser Line Width, MHz | Vacuum Chamber Input Power, mW | Lens Focus, m | Divergence, mm/m | Intensity, W/cm$^2$ |
|---|---|---|---|---|---|
| 422 | ~1 | 3 | 0.5 | 1.4 | 1.56 |
| 1092 | ~0.1 | 20 | 0.25 | 1 | 81.5 |
| 674 | ~1 | 6 | 0.4 | 1 | 9.55 |

Next, we analyzed the four-level system shown in Figure 3b, where the populations of the levels are designated as $N_1$, $N_2$, $N_3$, and $N_4$. The qualitative description of this system is implemented similarly to the previous case and formalized using a system of kinetic equations.

$$\frac{dN_1}{dt} = \frac{I_{422}\lambda_{422}^3}{2\pi hc}(N_2 - N_1) + \frac{I_{674}\lambda_{674}^3}{2\pi hc}(N_4 - N_1) + 2\pi\gamma_{422}N_2 + 2\pi\gamma_{674}N_4 \tag{4}$$

$$\frac{dN_2}{dt} = \frac{I_{422}\lambda_{422}^3}{2\pi hc}(N_1 - N_2) + \frac{I_{1092}\lambda_{1092}^3}{2\pi hc}(N_3 - N_2) - 2\pi(\gamma_{422} + \gamma_{1092})N_2 \tag{5}$$

$$\frac{dN_3}{dt} = \frac{I_{1092}\lambda_{1092}^3}{2\pi hc}(N_2 - N_3) + 2\pi\gamma_{1092}N_2 \tag{6}$$

$$\frac{dN_4}{dt} = \frac{I_{674}\lambda_{674}^3}{2\pi hc}(N_1 - N_4) - 2\pi\gamma_{674}N_4 \tag{7}$$

where $I_{422}$, $I_{674}$, and $I_{1092}$ are the radiation intensities for the wavelengths 422 nm, 674 nm, and 1092 nm (Table 1), respectively; $\lambda_{422}$, $\lambda_{674}$, and $\lambda_{1092}$ are the wavelengths 422 nm, 674 nm, and 1092 nm, respectively; $\gamma_{422}$, $\gamma_{674}$, and $\gamma_{1092}$ are the transition widths $5S_{1/2} - 5P_{1/2}$, $5S_{1/2} - 4D_{5/2}$, and $5P_{1/2} - 4D_{3/2}$, respectively. Similarly, we obtained the following population distribution: $N_1 \approx 0.2658253$, $N_2 \approx 0.234173$, $N_3 \approx 0.234175$, and $N_4 \approx 0.2658251$. From this it is clearly seen that at experimental radiation intensities, the population of the excited state $N_2$ decreases in the presence of radiation of 674 nm. This means that by observing the luminescence of the ion cloud at a wavelength of 422 nm, a decrease in the useful signal would be obtained. In the presence of 674 nm radiation, the population of the excited state is $N_2 \approx 0.234173$, and without it, it is $N_2 \approx 0.318957$, which corresponds to the luminescence drawdown of the ion cloud of $\approx 26.5\%$.

The unquestionably applied kinetic equations represent an idealized model that assumes the presence of laser radiation sources with infinitely narrow widths strictly located in resonances, which, in turn, is an unattainable condition under real experimental conditions. Also, this model does not take into account various mechanisms of line broadening, which significantly reduce the efficiency of the interaction of ions with laser radiation, and, accordingly, reduce the rate of output to stationary values. In addition, this model does not take into account quantum effects. Nevertheless, despite the above limitations, it provides a qualitative picture of what should be expected from the experiment of observing a decrease in the luminescence of an ion cloud at a wavelength of 422 nm.

## 4. Results and Widenings of 1092 nm Spectroscopy

With the laser radiation parameters described in the previous sections, it was planned to perform spectroscopy of the $5S_{1/2} - 4D_{5/2}$ clock transition. As a test of the trap operation and the compliance of the spectroscopy results with expectations, the theoretical estimates of the broadening of the repumping transition $4D_{3/2} - 5P_{1/2}$ were previously compared with the experimental results. Following the results of [30], in the cases of low and high laser radiation power ($\Omega \ll \delta$ or $\Omega \gg \delta$, where $\Omega$ is the Rabi frequency and $\delta$ is the frequency detuning), the pumping transition spectrum in the Lambda scheme [31] has a Lorentzein contour determined by the power broadening.

For stable operation of the Lambda laser-cooling scheme, a high-intensity repumping radiation is introduced into the capturing area, so that fluctuations of the laser frequency at a wavelength of 1092 nm do not affect the process. The corresponding width of the Lorentz contour can be estimated as

$$\Delta f_{FWHM_I} = \gamma\sqrt{\frac{I}{I_{sat}}} \tag{8}$$

where $\gamma$ is the natural linewidth, $I$ is the laser radiation intensity (calculated in Table 1 for all three transitions), and $I_{sat}$ is the saturation intensity, which is given by the expression:

$$I_{sat} = \frac{2\pi^2 hc\gamma}{3\lambda^3} \tag{9}$$

Substituting all the necessary known values in Formula (9), we obtained the following results in Table 2.

**Table 2.** Saturation intensities of the transitions used in this work.

| Wavelength, nm | Saturation Intensity, $\frac{mW}{cm^2}$ |
|:---:|:---:|
| 422 | $\approx 35$ |
| 1092 | $\approx 0.12$ |
| 674 | $\approx 1.7 \times 10^{-7}$ |

In the case of the repumping transition spectroscopy, the full width at half-height should be about 700 MHz.

The 1092 nm transition spectroscopy was performed at a constant cooling laser radiation frequency of 422 nm, equal to 710, 962.7 GHz. This corresponds to red detuning of the radiation frequency by $\approx 170$ MHz from the resonance. The choice of the detuning was made because of the following reasons: On the one hand, it allowed us to keep the cooling radiation red-detuned from the resonance despite the inaccuracy of the wavelength meter. On the other hand, it corresponds to a detuning of $\approx 8\gamma$, which still leads to effective Doppler cooling.

To investigate the repumping transition, we measured the dependence of the repumping efficiency on the repumping radiation frequency. For this, we scanned the laser radiation frequency over the resonance and measured the signal of the ion luminescence at the wavelength of 422 nm.

The resulting dependence of the luminescence signal was approximated by the Voigt profile (Table 3) and is shown in Figure 4. The full spectrum width at half-maximum was 710 MHz, which is in good agreement with the above estimate. The line center corresponds to the frequency of 274, 589.23 GHz and may be shifted due to coherent population trapping.

**Table 3.** Parameters of fitting by the Voigt profile of the repumping transition spectrum obtained from the luminescence signal.

| Wavelength, nm | Displacement of the Peak Center, MHz | Homogeneous Broadening, MHz | Inhomogeneous Broadening, MHz | FWHM, MHz |
|:---:|:---:|:---:|:---:|:---:|
| 1092 | 230 | 673.7 | 226.9 | 709.9 |

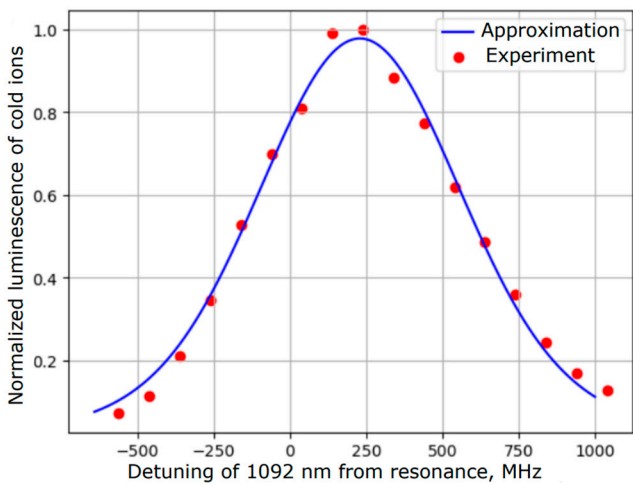

**Figure 4.** Spectroscopy of 1092 nm transition. The detuning was measured from the known literature resonance frequency of $\nu_0 = 274, 588.96$ GHz. The center of the peak corresponds to the frequency $\nu = 274, 589.19$ GHz, and the width at half-maximum is 710 MHz.

The problem of laser cooling in the $\lambda$-scheme (Figure 5) is coherent population trapping.

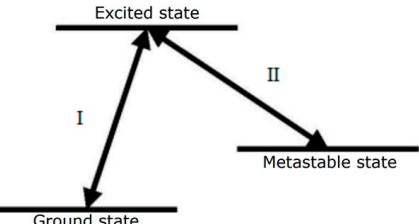

**Figure 5.** Three-level Lambda scheme. A characteristic feature of it is the presence of a long-lived metastable intermediate level.

The population of the excited state $\sigma_{22}$ as a function of the laser detuning from the resonances $\delta_I$ and $\delta_{II}$, the Rabi frequencies $\Omega_I$ and $\Omega_{II}$, and the transition widths $\gamma_I$ and $\gamma_{II}$ are succinctly described in [32] and presented as:

$$\sigma_{22} = N_{22}/D \tag{10}$$

$$N_{22} = 4\Omega_I^2\Omega_{II}^2(\gamma_I + \gamma_{II})(\delta_I - \delta_{II})^2 \tag{11}$$

$$
\begin{aligned}
D = (\delta_I - \delta_{II})^2 &\left\{ 8\Omega_I^2\Omega_{II}^2(\gamma_I + \gamma_{II}) + 16\Omega_I^2\gamma_{II}\left((\gamma_I + \gamma_{II})^2 + \delta_{II}^2\right)\right.\\
&\left. + 16\Omega_{II}^2\gamma_I\left((\gamma_I + \gamma_{II})^2 + \delta_I^2\right)\right\} + 8(\delta_I - \delta_{II})\left(\Omega_I^4\delta_{II}\gamma_{II} - \Omega_{II}^4\delta_I\gamma_I\right)\\
&+ \left(\Omega_I^2\gamma_{II} + \Omega_{II}^2\gamma_I\right)\left(\Omega_I^2 + \Omega_{II}^2\right)^2
\end{aligned}
\tag{12}
$$

$$\Omega = \gamma\sqrt{S_0/2} \tag{13}$$

Rabi frequencies were calculated from the saturation parameter by Formula (13). The population of the excited state in the $^{88}$Sr$^+$ ions were calculated by Formula (10) for various laser detuning from the resonance, with the parameters of the experimental setup, which is shown in Figure 6.

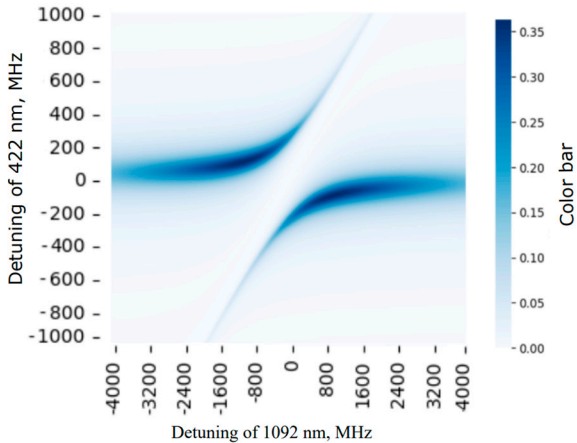

**Figure 6.** Population of the excited state for the $\Lambda-$system of $^{88}$Sr$^+$ ions, with saturation parameters corresponding to the experimental ones. Coherent population capture is observed at equal detuning of both lasers.

Having constructed the dependence of the excited state by Formula (10) on the detuning of 1092 nm, when the radiation frequency of 422 nm is red-detuned by 170 MHz from the resonance, a shift of 230 MHz to the right side of the resonance is clearly seen (Figure 7).

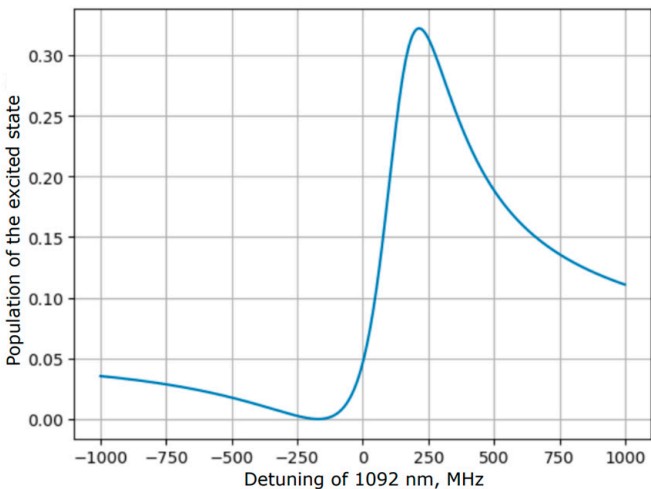

**Figure 7.** Population of the excited state for the $\Lambda$-system of $^{88}$Sr+ ions, with saturation parameters corresponding to experimental data. The line shows the dependence of the 1092 nm detuning on the resonance with a fixed detuning of 422 nm at 170 MHz in the red side from the resonance.

## 5. Spectroscopy of the 674 nm Clock Transition

The experiment on spectroscopy of the $5S_{1/2} - 4D_{5/2}$ clock transition based on the setup configuration described earlier was carried out according to two protocols. The main difference between them was as follows: the first protocol was implemented on a single-ion load, while the second protocol was re-loaded into the trap for each measurement. But in both cases, the useful signal was the luminescence of the ion cloud at a wavelength of 422 nm. Spectroscopy was based on the assumption that interaction with additional laser radiation, in this case 674 nm, leads to a mechanism for extracting ions from the cooling cycle, which results in a decrease in luminescence at a wavelength of 422 nm.

The first part of the experiment can be divided into two stages: preparation of the ion cloud for spectroscopy and spectroscopy itself. Ions were prepared for spectroscopy according to the following algorithm: (1) loading of ions with an energy of $\sim 10$ eV at a range of the variable component of the potential on the quadrupole of $V_a = 500$ V, and the 422 nm laser is stabilized at a frequency of 710, 962.5 GHz; (2) waiting for 10 s and decreasing $V_a = 350$ V; (3) waiting for 10 s and decreasing $V_a = 200$ V; (4) waiting for 10 s and reducing $V_a = 150$ V; (5) waiting for 10 s and tuning the 422 nm laser to a frequency of 710, 962.7 GHz for more deep cooling; and (6) waiting for 30 s for the ion luminescence to reach a stationary value. Next, the spectroscopy of the clock transition was performed: (1) accumulation of the cloud luminescence signal for 10 s; (2) opening of the shutter, blocking its optical path to laser radiation of 674 nm; (3) accumulation of the cloud luminescence signal for 10 s, but with additional radiation of 674 nm; (4) fixing the differences between the signals with the first and third steps; (5) closing the shutter, switching the 674 nm laser to 50 MHz and waiting for 30 s to reach the steady-state value of ion luminescence; and (6) repeating the algorithm from the first step.

According to the results of the numerical solution of the Bloch equations, the width of the clock transition spectrum is also determined by power broadening and the amplitude of the luminescence signal can reach half for a long interaction time. But the numerical solution did not take into account a number of broadenings.

Thus, according to our previous work with this experimental setup [29], the expected ion temperature ranges from fractions to tens of K, which corresponds to the Doppler broadening in the range from 5 to 200 MHz. The power broadening described in the previous section in the case of the clock transition is $\sim 70$ kHz, which is much smaller than the Doppler one. The broadening associated with the polarizability of the clock transition at the cooling transition frequency can also make a significant contribution. According to [32], the magnitude of the frequency shift from the power of laser radiation tuned to the cooling transition is determined by a linear coefficient, which, depending on the polarization of the

radiation, lies in the range from 23 to 45 $\frac{MHz}{W*cm^2}$. Strontium ions in the trap can be located in regions of different intensities, which can lead to inhomogeneous broadening up to 60 MHz when estimating the intensity of cooling radiation given earlier.

The resulting experimental signal dependence was approximated by the Voigt function (Table 4). The result is shown in Figure 8. The 0.25(4) GHz profile width is almost completely determined by the Lorentz component of the Voigt profile, i.e., by a certain uniform broadening. The line center corresponds to the frequency $\nu = 444,779.022(14)(90)$ GHz. Here, the statistical error of 14 MHz associated with the data approximation is indicated in parentheses, and the systematical error associated with the wavelength meter is 90 MHz. The obtained value of, taking into account the error, fully agrees with the frequency obtained earlier in [33], where the accuracy was at the level of 1 Hz.

**Table 4.** Parameters of fitting by the Voigt profile of the clock transition spectrum obtained from the luminescence signal (first protocol).

| Wavelength, nm | Displacement of the Peak Center, MHz | Homogeneous Broadening, MHz | Inhomogeneous Broadening, MHz | FWHM, MHz |
|---|---|---|---|---|
| 674 | −21.7 | 247 | <<1 | 247 |

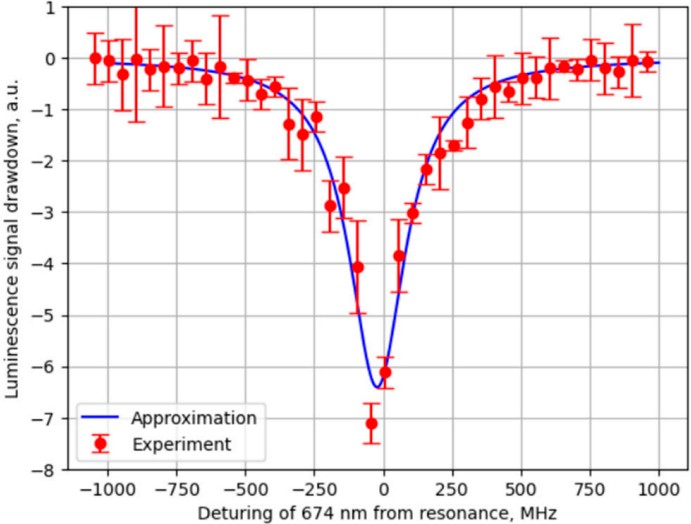

**Figure 8.** Spectroscopy of the $5S_{1/2} - 4D_{5/2}$ clock transition, where the vertical axis shows the decrease in the luminescence signal in the presence of radiation at 674 nm, the horizontal axis shows the radiation frequency in MHz, and the graph shows zero frequency $\nu_0 = 444,779$ GHz. The line center is at 22(14) MHz, and the width at half-maximum is 250(40) MHz.

The signal amplitude is 6%, which is lower than the theoretical estimate given above. However, the experimentally measured width of transition indicates the presence of other broadenings.

It is worth noting that the first protocol has a disadvantage due to the fact that the luminescence of the cloud decreases over time. The fact is that after the radiation of 674 nm is blocked, the luminescence of ions is not completely restored.

The second protocol was used in an attempt to overcome this problem, so it contains cloud preparation and spectroscopy algorithms similar to the first one, except for the fact that for each new measurement, the ion cloud was prepared anew. The resulting dependence is approximated by the Voigt function and is shown in Figure 9.

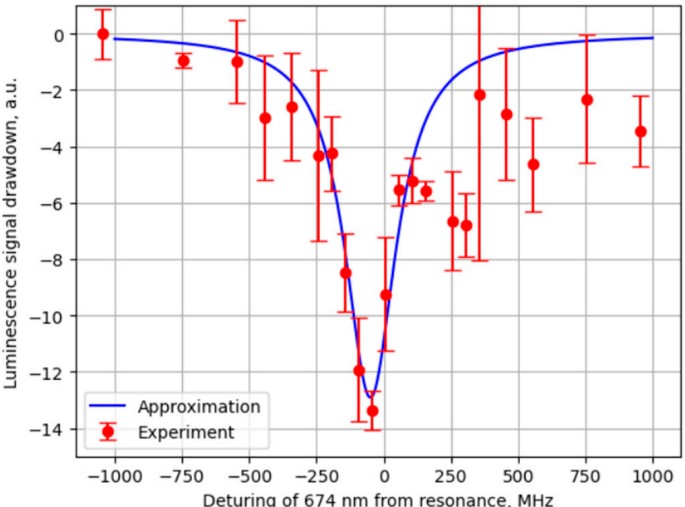

**Figure 9.** Spectroscopy of the $5S_{1/2} - 4D_{5/2}$ clock transition, where the vertical axis shows the difference between luminescences without and with radiation of 674 nm, summed to luminescence without 674 nm, and the horizontal axis shows the radiation frequency in MHz. Zero on the graph corresponds to the frequency $\nu_0 = 444,779$ GHz. The line center corresponds to the point $-12(8)$ MHz, and the width at half-height is $230(20)$ MHz.

Here, the spectrum width is $0.23(2)$ MHz. The center of the line in this case corresponds to the frequency $v = 444,778.988(8)(90)$ GHz. The second protocol is also not without its drawbacks due to the instability of the number of loaded ions from the ablation plasma, which generates an error and requires a larger number of measurements. But, in this case, a more pronounced signal is observed, a slightly narrower line, and the contour features that we associate with the imperfect experimental conditions become noticeable. We attributed the slight difference in the center frequency from the previous case to this, as well as to a possible drift in the values of the wavelength meter.

## 6. Consideration and Conclusions

In this paper, a qualitative theoretical and experimental possibility of detecting the interaction of 674 nm laser radiation with a clock quadrupole transition in high-energy $^{88}Sr^+$ ions obtained by laser ablation was demonstrated, and the results of the spectrometry of the clock and pump radiation are also presented. The parameters of the laser setup are presented, and various mechanisms causing line broadening are analyzed. In the case of a pumping laser, the width is almost entirely determined by the radiation intensity. A protocol, for which the clock transition spectroscopy was performed, is described.

The obtained experimental result, namely, the spectroscopy of the clock transition, agrees with previous experimental works [22,33]. Although our work does not provide more precise results, we showed that it is possible to detect the clock transition without the repump radiation at the wavelength of 1033 nm [34] in a hot cloud of ions.

**Author Contributions:** Conceptualization, E.T. and P.B.; methodology, E.T. and P.B.; software, E.T. and D.T.; validation, E.T., P.B., D.T., D.P., P.C. and K.T.; formal analysis, E.T., P.B. and D.T.; investigation, E.T., P.C. and K.T.; resources, P.B.; data curation, E.T., P.B., D.T., D.P., P.C. and K.T.; writing—original draft preparation, E.T., D.T., D.P., P.C. and K.T.; writing—review and editing, E.T., D.T. and D.P.; visualization, E.T. and K.T.; supervision, P.B.; project administration, P.B.; funding acquisition, P.B. All authors have read and agreed to the published version of the manuscript.

**Funding:** This research was funded by the Russian Science Foundation, grant number. 24-22-00068.

**Data Availability Statement:** No new data were created or analyzed in this study. Data sharing is not applicable to this article.

**Conflicts of Interest:** The authors declare no conflict of interest.

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
