# Peer review of "Registration of a Quadrupole Transition in High-Energy 88Sr+ Ions Obtained by Laser Ablation Method"

_photonics, doi:10.3390/photonics11040337_

Round 1

Reviewer 1 Report

Comments and Suggestions for Authors

Evgeniy Telnov at all shows in his paper the registration of a quadrupole transition in high-energy Sr+ ions. The ions are produced by laser ablation and loaded into a quadrupole Paul trap. The Sr+ ion is cooled down in this trap using laser cooling (422nm). The paper is well written, but I have some comments to make before I accept the paper for publication.

I want the authors to show how they load the trap. On one side they claim to have a vacuum better 10(-9) mbar. On the other side they do laser ablation to get their Sr+ ions in the trap. Maybe they can show a picture of how they do it.

Figure 2 is so not clear to me. The flashes should clearly indicate the transitions.

Figure 9 The authors claim that the centerline corresponds to 𝑣=444778.988(8)(90)𝐺𝐻𝑧. On the right side of the spectra, I see a couple of 260 Mhz and a second line at about 520 Mhz. Where do these lines come from? (They have about the same line with ?)

I like to ask the authors to show error bars  in Figure 8 and Figure 9

Author Response

.

Reviewer 2 Report

Comments and Suggestions for Authors

Comments on the Quality of English Language

It is necessary to check the language and the usage of the symbols, like some unit, such as MHz, GHz, etc.

Author Response

.

Reviewer 3 Report

Comments and Suggestions for Authors

Excellent presentation. I have no recommendations.

Author Response

.

Round 2

Reviewer 2 Report

Comments and Suggestions for Authors

The authors answered all the questions and revised the manuscript. It can be accepted now.

Comments on the Quality of English Language

No.